# Evaluation of shotgun metagenomics as a diagnostic tool for infectious gastroenteritis

Kjersti Haugum[1,2]*, Anuradha Ravi[3¤], Jan Egil Afset[1,2], Christina Gabrielsen Ås[1,2]

1 Department of Medical Microbiology, St. Olavs Hospital, Trondheim University Hospital, Trondheim, Norway, 2 Department of Clinical and Molecular Medicine, Faculty of Medicine and Health Sciences, NTNU, Trondheim, Norway, 3 Department of Circulation and Medical Imaging, Faculty of Medicine and Health Sciences, Norwegian University of Science and Technology, Trondheim, Norway,

¤ Current address: Department of Medical Genetics, St. Olavs hospital, Trondheim University Hospital, Trondheim, Norway
* Kjersti.haugum@stolav.no

## Abstract

Infectious gastroenteritis is a significant health issue globally. Identifying the causative pathogen is crucial for treatment, infection control and epidemiological surveillance. While PCR-based analyses are fast and sensitive, they only detect known pathogens. Clinical metagenomics can potentially identify novel or unexpected pathogens. This study aimed to evaluate shotgun metagenomics for detecting diarrhoeal pathogens in faecal samples from patients with infectious gastroenteritis and spiked samples from healthy donors, compared to PCR. DNA from clinical faecal samples (n = 12), spiked samples (n = 36), and control samples (n = 7) were analysed by PCR and shotgun metagenomics sequencing. Reads were taxonomically assigned, assembled, and binned into MAGs. MAGs were taxonomically assigned, and virulence genes were detected in bacterial assemblies and MAGs. Pathogens detected by PCR were also identified by taxonomic assignment of reads, though with lower sensitivity. Taxonomic assignment of MAGs identified 50% of bacterial pathogens and HAdV-F. Additional potential pathogens were observed in most samples. More bacterial virulence genes were detected in assemblies than in MAGs. In spiked samples, *C. jejuni* and HAdV-F were detected by both PCR and metagenomics, with significant correlation between Cq values and reads. Parasites were detected by few reads. Metagenomics has lower sensitivity compared to PCR but can provide supplementary information relevant for treatment. Challenges include additional potential pathogens, background microbiome, and introduced kitome, necessitating optimized extraction methods and strict quality controls.

**Data availability statement:** Sequence related files are available from the NCBI database (https://www.ncbi.nlm.nih.gov/bioproject/?term=PRJNA1218764). All other relevant data are available within the paper and its Supporting Information files.

**Funding:** This study was funded by grant 14/8337-124/NISLIN from St. Olavs hospital (https://www.stolav.no/) received by JEA. The funders had no role in study design, data collection and analysis, decision to publish, or preparation of the manuscript.

**Competing interests:** The authors have declared that no competing interests exist.

## Introduction

Infectious gastroenteritis is a leading cause of diarrhoea globally, with a need for rapid and accurate diagnostics for patient care and treatment. Worldwide, diarrhoea still has a high disease burden and was in 2019 regarded as the fifth leading cause of disability-adjusted life-years (DALYs) overall [1]. Among children less than 5 years old, diarrhoea was the third leading cause of DALYs, with rotavirus infection as the main aetiology [1,2].

Detection of pathogenic microbes causing diarrhoea has traditionally been based on microscopy and culturing. In recent years, diagnostic laboratories have increasingly implemented PCR-methods, which have proven successful and cost effective for the detection of pathogens that are not readily culturable. However, even though these methods are rapid and have shown high sensitivity, they can only detect the pathogens they target [3]. Since there is a plethora of potential enteric pathogens, even a large panel of PCR analyses including the new syndromic panels, will not cover all potential pathogens.

Shotgun metagenomics has emerged as a promising tool in the diagnosis of infectious pathogens, including gastrointestinal pathogens [4–8]. An advantage of this approach is the potential for providing information of the whole genome of all organisms present in a sample, as well as providing information on virulence and resistance genes [9,10]. A faecal sample is a complex matrix, in which normal faecal microbiota and host cell debris are represented alongside potential pathogenic microbes [11]. The number/amount of host cells and non-pathogenic microbes present in faecal sample compared to the amount of pathogenic microbes will potentially influence on the sensitivity and quality of a metagenomics test. In addition, other factors like extraction protocol, library preparation and sequencing protocol, as well as data analysis tools will also influence on the quality and sensitivity of the assay. Despite potential difficulties, shotgun metagenomics could add knowledge in diagnostics of gastroenteritis, as reports estimate that the etiological agent is unknown in approximately 40% of cases with gastroenteritis using current methods [12].

In this study, we aimed to use shotgun metagenomics and bioinformatics analyses to detect bacterial, viral and protozoal pathogens in faecal samples from patients with infectious gastroenteritis and spiked faecal samples from healthy donors, compared with standard laboratory methods.

## Materials and methods

### Clinical faecal samples and clinical routine diagnostics

We retrospectively selected 12 faecal samples from patients with acute gastroenteritis, having positive PCR results for *Campylobacter* spp. (n = 3), *Clostridioides difficile* toxin B (n = 1), *Salmonella* spp. (n = 2), Shiga toxin-producing *Escherichia coli* (STEC) (n = 2), *Giardia intestinalis* (n = 1), *Cryptosporidium* (n = 1), and Human mastadenovirus F (HAdV-F) (n = 2). The samples were collected in sterile collection containers (International Scientific Supplies Ltd, United Kingdom) from August to October 2016, and were stored at −20°C until DNA extraction was performed. Before nucleic acid

extraction for PCR analyses, a pea-sized amount of each faecal sample was incubated overnight in Difco selenite broth (BD Life Sciences, USA) for enrichment of *Salmonella* spp. To enrich for parasites, another pea-sized amount of the faecal samples was frozen overnight in a solution of 200 µL molecular grade water and 200 µL Nuclisens easyMAG lysis buffer (bioMeriéux, France). Subsequently, 200 µL thawed supernatant from the frozen samples was mixed with 50 µL selenite broth before extraction of nucleic acids using NucliSens easyMAG (bioMeriéux, France). Inclusion of samples was based on positive PCR using Allplex™ GI-Bacteria(I) Assay, Allplex™ GI-Bacteria(II) Assay and Allplex™ GI-Virus Assay (Seegene, Republic of Korea) for detection of bacterial and viral pathogens, or RIDA®GENE Parasitic Stool Panel (R-Biopharm AG, Germany) for parasite detection. The sequencing workflow for clinical samples with downstream laboratory and bioinformatics analyses are summarised in Fig 1.

### Spiked faecal samples

Faecal samples from four healthy donors, denoted BP1, BP2, BP4 and BP5 were collected in sterile containers. From each of the four samples, we prepared a liquid and homogenous subsample by mixing with sterile phosphate buffered saline (PBS). All subsamples were spiked with *C. jejuni*, *G. intestinalis* and HAdV-F, representing common gastrointestinal bacterial, parasitic and viral pathogens respectively (Fig 1). For spiking with *C. jejuni*, we used a liquid culture of strain ATCC 33252 with starting concentration $2.0 \times 10^8$ CFU/mL (colony forming units per mL). In this study, it was not possible to obtain pure cultures of *G. intestinalis* and HAdV-F, and thus for spiking we used two different clinical faecal samples PCR positive for *G. intestinalis* (original Cq-value 25) and HAdV-F (original Cq-value 8) respectively.

The liquid subsamples were taken for downstream analysis as follows; All were separately spiked to a final volume of 1 mL with 100 µL of a liquid culture of *C. jejuni* ATCC 33252 with concentration $2.0 \times 10^8$ CFU/mL, yielding the final concentration $2.0 \times 10^7$ CFU/mL, and 100 µL each of the faecal samples PCR positive for *G. intestinalis* and HAdV-F. This dilution was defined as $10^{-1}$, and from this dilution we prepared technical triplicates for each of the four spiked subsamples. These were thereafter diluted ten-fold in corresponding liquefied faeces, from $10^{-1}$ to $10^{-5}$. For the following DNA extraction, PCR analyses and shotgun sequencing, we used the $10^{-1}$, $10^{-3}$, and $10^{-5}$ dilutions of the triplicate spike-in samples, resulting in 36 samples. In addition, 1 mL from each of the non-spiked subsamples were collected and included in the same analyses, as non-spiked negative controls (BP_Neg, n = 4).

### DNA isolation for whole genome metagenomics sequencing

DNA from clinical faecal samples (n = 12), spiked faecal samples (n = 36) and faecal samples from healthy donors without spiked pathogens (n = 4) was isolated with the QIAamp DNA Stool Kit (Qiagen, Germany) according to the protocol "Qiagen + Bead Beating (QIAStool+BB)" [13] with the following modifications: For DNA extraction, 200 µl, or alternatively, 200 mg faecal sample was mixed with 1400 µl ASL buffer (from the QIAamp DNA Stool Kit) in a Lysing Matrix A tube (MP Biomedicals LCC, USA). The samples were vortexed and then homogenized three times for 30 s at speed 6.0 using a FastPrep®-24 Instrument (MP Biomedicals LCC, USA). The samples were placed on ice between each bead-beating step. The samples were then heated for 15 min at 95°C before the remaining protocol was performed according to manufacturer's instructions using a QIAcube (Qiagen, Germany). DNA was eluted in 200 µl Buffer AE. DNA concentrations of all samples were measured using a Qubit® Fluorometer and Qubit™ dsDNA HS Assay Kit (Thermo Fisher Scientific, USA). In addition, DNA concentration and $A_{260}/A_{280}$ and $A_{260}/A_{230}$ ratios was measured for all samples with NanoDrop (Thermo Fisher Scientific, USA). As negative controls, lysis buffer (ASL), molecular grade water (MGW) and Phosphate Buffered Saline (PBS) were included in the DNA extraction and all subsequent steps (n = 3).

### PCR analyses of spiked faecal samples from healthy donors

For spiked faecal samples, *C. jejuni* was detected using a real-time PCR targeting the *mapA* gene [14]. The PCR mixture contained Custom Multiplex PCR SuperMix, UNG (QuantaBio, USA), 300 nM forward primer, 300 nM reverse primer,

Sample preparation, PCR and library preparation

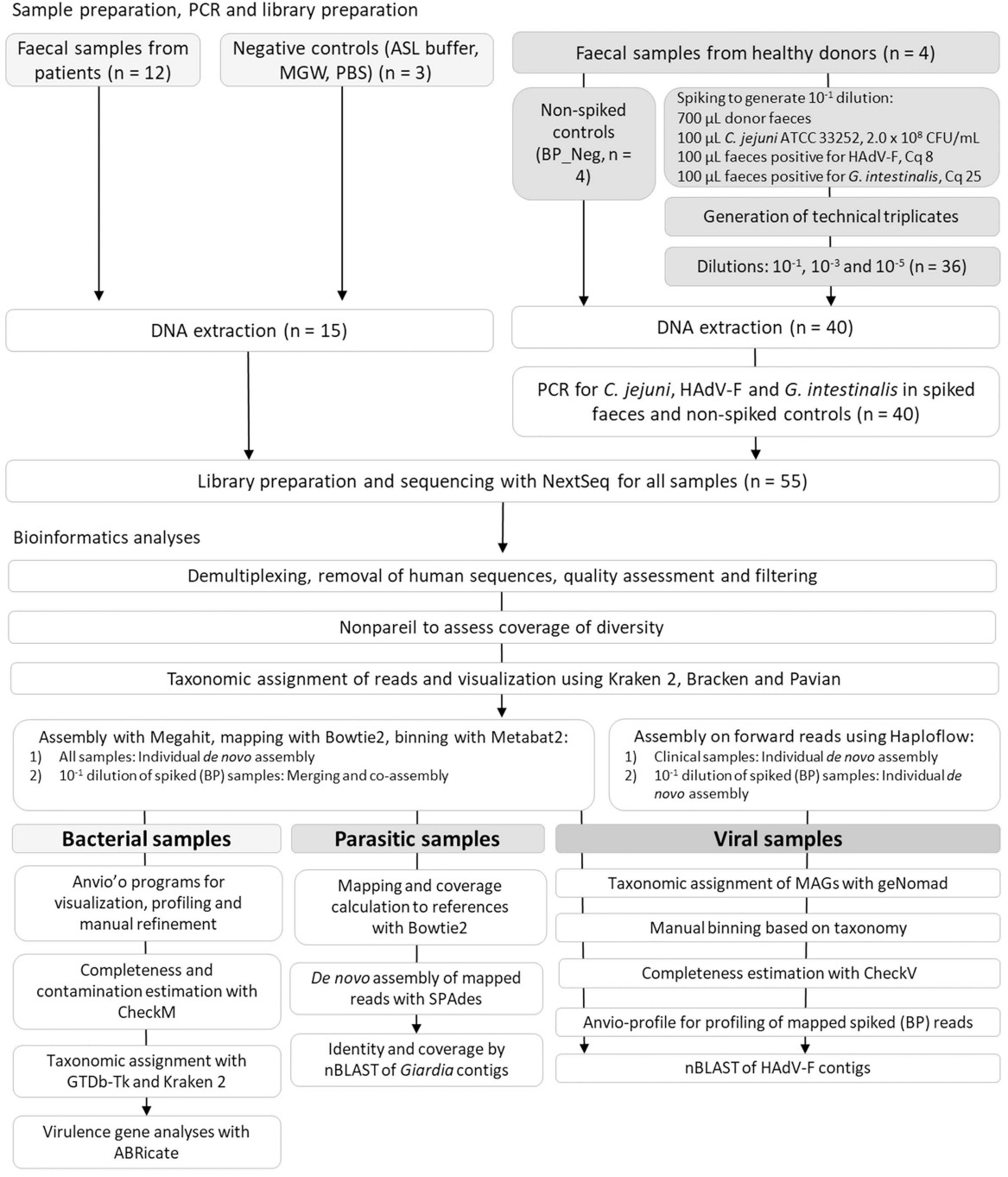

**Fig 1. Overview of the processes for sample preparation, PCR and library preparation, sequencing and bioinformatics analyses for the feacal samles analysed in this study.**

200 nM probe (TIB Molbiol Syntheselabor GmbH, Germany), and 5 µl of extracted DNA, in a total of 20 µl. The temperature profile was as follows: 45°C for 5 min, 95°C for 3 min followed by 40 cycles of 95°C for 10 sec and 55°C for 30 sec. DNA from *C. jejuni* ATCC 33252 was used as positive control in the PCR reaction. HAdV-F was detected by real-time

PCR targeting the hexon gene [15,16]. The PCR mixture contained PerfeCTa SYBR Green FastMix (QuantaBio, USA), 600 nM each of forward and reverse primers (TIB Molbiol Syntheselabor GmbH, Germany), and 5 µl of extracted DNA in a total of 20 µl. Here the temperature profile was 95°C 3 min, 40 cycles of 95°C for 10 sec, 56°C for 15 sec and 72°C for 20 sek. DNA from ATCC cat.no.VR-1 adenovirus1 was used as positive control. *Giardia intestinalis* was detected with the RIDA®GENE Parasitic Stool Panel (R-Biopharm AG, Germany), using Vircell Amplirun Giardia DNA control (Vircell Micro-biologists, Spain) as positive control for *Giardia intestinalis*. The faecal samples from healthy donors without any addition of pathogens, as well as molecular grade water, were included as negative controls in all PCR reactions.

## Shotgun sequencing

Sequencing libraries for all samples including controls (n = 55) were prepared using the Nextera® XT DNA Sample Preparation Kit (Illumina, USA) according to the manufacturer's instruction. Libraries were paired-end sequenced on the NextSeq system using three NextSeq 550 System High-Output Kits (Illumina) for 150 x 2 cycles. The sequencing service was provided by the Genomics Core Facility (GCF) at the Norwegian University of Science and Technology (NTNU).

## Bioinformatics analysis

**Sequence filtering and quality control.** The bioinformatic analyses were performed using default options unless otherwise specified. Sequencing reads were demultiplexed using bcl2fastq v2.18.0.12 (Illumina, USA). The resulting fastq files were aligned to GRCh38 using bwa v0.7.17 [17] for removal of human sequences. The quality of the sequences was assessed using FastQC v0.11.5 [18]. Quality filtering was done using Trimmomatic v0.32 [19] to remove Illumina adapters, leading and trailing sequences below Q30, and reads below 36 bp (Fig 1). The Nonpareil package [20] and RStudio (RStudio Team (2021). RStudio: Integrated Development Environment for R. RStudio, PBC, Boston, MA URL http://www.rstudio.com/) was used to assess coverage of diversity.

**Taxonomic analyses of metagenomics reads.** Taxonomic assignment of trimmed reads was performed using Kraken 2 v2.1.1 [21] with the Kraken 2 standard database updated June 7th 2022 (https://benlangmead.github.io/aws-indexes/k2). The database consists of taxonomic information and complete genomes in RefSeq from archaea, bacteria, fungi, viruses, protozoa, known vectors (UniVec_Core) and human sequences. In addition, Bracken (Bayesian Reestimation of Abundance with Kraken) [22] was used to convert counts into relative abundances. KrakenTools (https://github.com/jenniferlu717) was used to generate the Kraken report. Only reads assigned to pathogens that were searched for in our routine diagnostics were included for further analysis. This included *Aeromonas* spp., *Campylobacter coli, C. jejuni, Clostridioides difficile,* Shiga toxin-producing *E. coli* (STEC), enteropathogenic *E. coli* (EPEC), enterotoxigenic *E. coli* (ETEC), enteroaggregative *E. coli* (EAEC), *Salmonella* spp.*, Shigella* spp./enteroinvasive *E. coli* (EIEC), *Vibrio* spp., *Yersinia enterocolitica*, *Cryptosporidium* spp.*, Giardia intestinalis,* and Human Mastadenovirus F (HAdV-F). To control for potential taxonomic misclassifications by Kraken 2, KrakenUniq v1.0.4 [23,24] was additionally run on the negative control samples and the non-spiked negative controls (BP_Neg).

**De novo assembly.** Quality controlled reads for each sample were de novo assembled using Megahit v1.1.3 [25–27]. Additionally, reads from the $10^{-1}$ dilution of the BP samples from healthy donors (all samples and replicates) were merged and co-assembled. This was done to generate a single reference assembly, and to improve genome assembly by increasing read depth. Reads from clinical and spiked samples were mapped back to the assembled contigs using Bowtie2 v2.3.4.1 [28] and SAMtools v1.7 [29]. Reads from the spiked samples were also mapped to the co-assemblies. Forward reads from clinical viral samples and the $10^{-1}$ dilutions of the BP samples individually were additionally de novo assembled using Haploflow v1.0 [30].

**Metagenomic binning and taxonomic assignment of metagenome-assembled genomes (MAGs).** Metabat2 version 2.12.1 [31–33] was used to bin contigs into probable genomes. For bacterial samples, all metagenome-assembled genomes (MAGs) were imported into anvi'o version 6.2 [34]. The programs anvi-interactive were used to visualise

genome bins, anvi-profile to estimate coverage and detection of the MAG statistics for each sample, and anvi-refine to manually refine the genomes. CheckM version 1.1.2 [35] was used to estimate the completeness and contamination of the MAGs. To analyse the closest taxonomic representation of the bacterial MAGs, Genome Taxonomy database Toolkit (GTDb-Tk) version 0.3.2 [36] with GTDb database release95 (updated 2021) was used. In addition, for specific bins in which target pathogens were detected, taxonomic assignment were also done on contigs using Kraken 2.

For viral samples, geNomad v1.3.3 [37] was used for taxonomic assignment of viral contigs. Taxonomically related contigs were manually extracted into bins. CheckV version 1.0.1 [38] was used to estimate the completeness of the MAGs. The contigs were mapped to reference Hannover2022 (ON815883.1), and anvio-profile was used to estimate coverage and detection. Lastly, the contigs were subjected to nucleotide BLAST using the NCBI non-redundant nucleotide (nr/nt) database to identify percentage identity and percentage coverage against the most closely related genomes.

**Species-specific detection.** *Giardia intestinalis*

A reference mapping approach was used to identify Giardia spp. related reads in both spiked samples and PCR-positive clinical samples. The metagenomic reads were mapped against the reference genomes of *G. intestinalis* (GCA_000002435.2), *G. muris* (GCA_006247105.1) and *G. lamblia (intestinalis) (*GCA_000182665.1) using Bowtie2 v2.3.4.1 and SAMtools v1.7. Mapped reads were visualised in Qualimap2 [39]. Reads that mapped to *Giardia* spp. reference genomes were extracted from BAM files using Bedtools v2.26.0 [40] bamtofastq feature and assembled into contigs using SPAdes v3.15.3 [41]. Lastly, the contigs were subjected to nucleotide BLAST using the NCBI non-redundant nucleotide (nr/nt) database to identify percentage identity and alignment length against the most closely related genomes. *Campylobacter jejuni* and *Clostridioides difficile*

By taxonomic assignment using Kraken 2 and KrakenUniq, reads from *C. jejuni* and *C. difficile* were detected in all BP_Neg samples from healthy donors without spiked pathogens. The reads from these specific pathogens were extracted using KrakenTools and assembled with SPAdes (with meta option), and the ten largest contigs from each pathogen were manually inspected by nucleotide BLAST using the NCBI non-redundant nucleotide (nr/nt) database to identify identity against the most closely related genomes. Results were based on the annotation from the BLAST search.

**Virulence gene analyses.** By taxonomic assignment of reads by Kraken 2, bacterial virulence genes, with focus on the genes most relevant to the target bacterial pathogens, were identified in assembled contigs and MAGs from bacterial clinical samples using ABRicate v1.0.1 (https://github.com/tseemann/abricate) with Virulence Factor Database (VFDb) [42]. Default parameters were used and only genes that had > 95% alignment length to the reference genes were considered.

**Statistical analyses.** To evaluate correlation of Cq values against read counts for each dilution of the spiked samples, Pearson's Product-Moment Correlation was calculated in R using ggpubr v0.6.0 library.

**Ethics statement.** This study was approved by the Regional Committee for Medical and Health Research Ethics, REC Central (REC number 2015/207). Faecal samples sent to the medical microbiology department for routine diagnostics in the period 1st August to 1st October 2016 were retrospectively selected for inclusion in the study. Written informed consent was obtained from all participants included in the study. Participants were eligible for inclusion if they were more than 18 years old. After data collection, all analyses were performed on de-identified data.

## Results

### Sequencing output and quality metrics

In total 55 samples, including clinical faecal samples, faecal samples from healthy donors (spiked and non-spiked), and negative controls were sequenced using a shotgun metagenomics approach in this study (Fig 1). Sequencing of the clinical samples resulted in an average 53.4 ± 21.1 million (M) paired-end reads per sample after quality control filtering, where on average 0.16 ± 0.19% of reads did not pass quality filtering (S1 Fig). The samples from healthy donors (spiked and non-spiked) resulted in 45.5 ± 12.2 M paired-end reads per sample after quality filtering, with average loss of 0.10 ± 0.03% reads. The metrics for the negative controls were 0.20 ± 0.16 M paired-end reads per sample, with average

loss of 6.37 ± 3.01% reads. The calculated Nonpareil diversity for the clinical samples was in the range 15.3–18.7, and for the spiked samples (BP1, BP2, BP4 and BP5) in the range 17.6–18.6 (S1 Table). The abundance-weighted average coverage of the clinical samples ranged from 0.883 to 0.995, where 1 would indicate complete coverage of diversity. The abundance-weighted average coverage for the spiked samples was 0.89–0.95. The number of classified and unclassified reads by Kraken were in average 19.85 ± 0.84 M and 2.47 ± 1.96 M reads (S2 Table), respectively, while human contamination in clinical samples and BP samples after mapping were on average 0.00028%.

### Detection of target pathogens by PCR and taxonomic assignment of reads

**Clinical samples.** For the clinical faecal samples, all pathogens detected by PCR-based diagnostics were also identified by taxonomic assignment of metagenomics reads, including bacterial, viral and parasitic pathogens. The detected number of target pathogen-associated reads was highly variable between samples (Table 1, S2 Table). By PCR, *Campylobacter spp.* was detected in Sample 2 with 989 (0.003%) reads, and in Samples 5 and 6 with 82,347 (0.237%) and 70,272 (0.295%) reads, respectively. Bracken analysis showed that for Samples 2 and 5, more reads were associated to *Campylobacter jejuni* than to *C. coli*, while the opposite was observed in Sample 6. For Sample 4 diagnosed with *Clostridioides difficile* having toxin B by PCR, 26,112 (0.116%) reads were associated to *C. difficile*. Samples 11 and 13 which were both diagnosed as Shiga toxin-producing Escherichia coli (STEC) by PCR, had more than 2 M reads associated to *E. coli*. By PCR, *Salmonalla enterica* was detected in Sample 12 and Sample 15. Sample 12 contained > 2 M reads (14.120%) associated to *S. enterica*, while Sample 15 contained 925 reads (0.006%) associated to *S. enterica*.

The clinical samples diagnosed with parasites by PCR showed very few reads associated to parasites (Table 1b); Sample 10 had 104 reads classified to *Giardia intestinalis* and sample 14 had 298 reads to *Cryptosporidium* spp. No reads from parasites were observed in the other samples. In clinical samples diagnosed with Human adenovirus F, the virus was identified in Sample 3 with 2,7 M reads (9.667%) and in Sample 7 with > 460,000 reads (2.676%). No other samples contained reads associated to HAdV-F or other adenoviruses. Several samples showed false positive results using the taxonomic assignment approach, with high abundances of *E. coli*. Additionally, *C. difficile* was detected in all clinical samples (Table 1, Table 2). Of the negative controls, the lysis buffer and phosphate buffered saline (PBS) had reads assigned to *E. coli*, observed with both Kraken 2 and KrakenUniq. Otherwise no reads were associated to the pathogens screened for in routine diagnostics in the negative controls.

**Spiked samples.** *C. jejuni* and HAdV-F were detected in several dilutions by both PCR and sequencing. For *C. jejuni* and HAdV-F, PCR of the spiked faecal samples from healthy volunteers (denoted BP) showed that the Cq values were lowest in the $10^{-1}$ dilution and as expected increasing with approximately six Cq values between the $10^{-1}$, $10^{-3}$ and $10^{-5}$ dilutions (Table 3). Within each dilution, the Cq values varied only slightly between technical replicates as well as between BP samples. By sequencing, the highest number of reads for *C. jejuni* and HAdV-F were were found in the $10^{-1}$ dilution, with average number of reads of 15,648.2 and 30,806.7, respectively. In the $10^{-3}$ and $10^{-5}$ dilutions, the average number of reads for *C. jejuni* were 4,341.4 and 3,821.8 respectively, almost similar to the 3,965.8 reads of *C. jejuni* observed in BP_neg (non-spiked) samples supposed not to be infected with *C. jejuni*. The BP_neg sample was negative for *C. jejuni* by PCR. For HAdV-F, only 276 ± 181 reads on average were observed in the $10^{-3}$ dilution, while HAdV-F-associated reads not were detected in the $10^{-5}$ dilution in any of the HAdv-F-spiked samples. A small number of HAdVF reads were found in the non-spiked BP5-sample (Table 3). The calculated Pearson's Product-Moment Correlation of Cq values against read counts for the $10^{-1}$, $10^{-3}$ and $10^{-5}$ dilutions of *C. jejuni* showed low but statistically significant correlation (R = −0.69, p = 4.317e-06). Also for HAdV-F, significant correlation between Cq values and reads was observed (R = −0.82, p = 9.79e-07). In three out of four BP samples, in both spiked and non-spiked BP samples, a high number of *E. coli* reads were observed. *C. difficile* reads were also observed in all BP samples using both Kraken 2 and KrakenUniq. Extraction and assembly of *C. jejuni* and *C. difficile* reads from BP_neg samples, followed by manual BLAST analysis showed that the majority of the resulting contigs had closest match to functions on mobile genetic elements, e.g., plasmids and

**Table 1.** (a) Number of metagenomics reads and percent abundance by taxonomic assignment using Kraken 2 in clinical bacterial faecal samples. Only reads assigned to bacterial pathogens searched for in routine diagnostics were included. (b) Taxonomic assignment of metagenomics reads by Kraken 2 in clinical faecal samples where parasitic and viral pathogens were detected by routine PCR. Results of taxonomic assignment of reads from negative controls.

| Pathogen detected by sequencing | (a) No. of reads (%) | | | | | | | |
|---|---|---|---|---|---|---|---|---|
| Sample No – pathogen detected by routine PCR (Cq value) | Sample 2 – *Campylobacter* spp. (23) | Sample 4 – *Clostridioides difficile* (27) | Sample 5 – *Campylobacter* spp. (22) | Sample 6 – *Campylobacter* spp. (26) | Sample 11 – *Escherichia coli* (Shiga toxin-producing) (22 (*eae*), 34 (*stx1*)) | Sample 12 – *Salmonella* spp. (27) | Sample 13 – *Escherichia coli* (Shiga toxin-producing) (21 (*stx1*)) | Sample 15 – *Salmonella* spp. (24) |
| *Aeromonas* spp.* | 0 | 0 | 0 | 0 | 0 | 1,511,974 (9.060) | 0 | 0 |
| *Campylobacter coli* | 294 (0.001) | 0 | 15 (0.00004) | 70,272 (0. 295) | 0 | 0 | 59 (0.001) | 199 (0.001) |
| *Campylobacter jejuni* | 989 (0.003) | 92 (0.0004) | 82,347 (0.237) | 2,562 (0.011) | 0 | 0 | 861 (0.010) | 3,005 (0.020) |
| *Clostridioides difficile* | 7,555 (0.021) | 26,112 (0.116) | 5,042 (0.014) | 2,329 (0.010) | 8,658 (0.030) | 104 (0.001) | 1,743 (0.019) | 4,007 (0.027) |
| *Escherichia coli* | 217 (0.001) | 1,038 941 (4.626) | 50,368 (0.145) | 5,292,311 (22.250) | 2,361,637 (6.750) | 7,532,243 (45.120) | 2,320,698 (25.260) | 1,544,410 (10.272) |
| *Salmonella enterica* | 0 | 2,252 (0.010) | 0 | 15,071 (0.060) | 1,889 (0.010) | 2,356,721 (14.120) | 2,813 (0.031) | 925 (0.006) |
| *Salmonella* spp., excluding *S. enterica* | 0 | 0 | 0 | 0 | 0 | 2 712 (0.020) | 0 | 0 |
| *Shigella* spp./ Enteroinvasive *E. coli* (EIEC) | 0 | 921 (0.004) | 0 | 107,069 (0.450) | 4,619 (0.010) | 25,899 (0.160) | 9,495 (0.103) | 2,228 (0.015) |
| *Yersinia enterocolitica* | 0 | 0 | 0 | 0 | 129 (0.0004) | 0 | 0 | 0 |
| *Cryptosporidium* spp. | 0 | 0 | 0 | 0 | 0 | 0 | 0 | 0 |
| *Giardia intestinalis* | 0 | 0 | 0 | 0 | 0 | 0 | 0 | 0 |
| Human Mastadeno-virus F | 0 | 0 | 12 (0.00003) | 0 | 0 | 0 | 0 | 0 |
| **Total No. of classified reads from Bracken** | **36,614,646** | **22,460,395** | **34,814,748** | **23,784,334** | **35,005,454** | **16,695,051** | **9,188,873** | **15,035,219** |

| Pathogen detected by sequencing | (b) No. of reads (%) | | | | | | |
|---|---|---|---|---|---|---|---|
| Sample No – pathogen detected by routine PCR (Cq value) | Sample 10 – *Giardia intestinalis* (25) | Sample 14 – *Cryptosporidium* spp. (31) | Sample 3 – Human mastadeno-virus F (8) | Sample 7 – Human mastadeno-virus F (9) | Lysis buffer – NA** | Molecular grade water – NA | Phosphate buffered saline – NA |
| *Aeromonas* spp.* | 0 | 0 | 0 | 0 | 0 | 0 | 0 |
| *Campylobacter coli* | 855 (0.010) | 1,880 (0.010) | 66 (0.0002) | 43 (0.0002) | 0 | 0 | 0 |
| *Campylobacter jejuni* | 5,411 (0.035) | 21,085 (0.114) | 71 (0.0003) | 378 (0.002) | 0 | 0 | 0 |
| *Clostridioides difficile* | 8,480 (0.060) | 24,594 (0.133) | 3,115 (0.011) | 23,251 (0.133) | 0 | 0 | 0 |
| *Escherichia coli* | 1 124 (0.055) | 185,524 (1.006) | 708,980 (2.537) | 1,958 (0.011) | 1,622 (1.676) | 0 | 997 (6.008) |
| *Salmonella enterica* | 0 | 0 | 3,876 (0.014) | 0 | 0 | 0 | 0 |

*(Continued)*

**Table 1.** (Continued)

| Pathogen detected by sequencing | (a) No. of reads (%) | | | | | | | |
|---|---|---|---|---|---|---|---|---|
| Sample No – pathogen detected by routine PCR (Cq value) | Sample 2 – *Campylobacter* spp. (23) | Sample 4 – *Clostridioides difficile* (27) | Sample 5 – *Campylobacter* spp. (22) | Sample 6 – *Campylobacter* spp. (26) | Sample 11 – *Escherichia coli* (Shiga toxin-producing) (22 (*eae*), 34 (*stx1*)) | Sample 12 – *Salmonella* spp. (27) | Sample 13 – *Escherichia coli* (Shiga toxin-producing) (21 (*stx1*)) | Sample 15 – *Salmonella* spp. (24) |
| *Salmonella* spp., excluding *S. enterica* | 0 | 0 | 0 | 0 | 0 | 0 | 0 | |
| *Shigella* spp./ Enteroinvasive *E. coli* (EIEC) | 0 | 1,942 (0.011) | 11,362 (0.041) | 0 | 0 | 0 | 0 | |
| *Yersinia enterocolitica* | 0 | 0 | 0 | 0 | 0 | 0 | 0 | |
| *Cryptosporidium* spp. | 0 | 298 (0.002) | 0 | 0 | 0 | 0 | 0 | |
| *Giardia intestinalis* | 104 (0.001) | 0 | 0 | 0 | 0 | 0 | 0 | |
| Human Mastadenovirus F | 0 | 0 | 2,701 445 (9.667) | 467,597 (2.676) | 0 | 0 | 0 | |
| **Total No. of classified reads from Bracken** | **15,425,382** | **18,442,806** | **27,946,183** | **17,473,646** | **96,759** | **3,926** | **16,594** | |

*\*Aeromonas* spp. regarded as potential human pathogens are usually *A. hydrophila*, *A. caviae*, and *A. veronii* biovar *sobria*.

\*\*NA: not applicable

Because there were no hits on *Vibrio* spp. from Kraken and Bracken, *Vibrio* spp. are not included in the table.

**Table 2. Number of true positive, true negative, false positive and false negative samples for the taxonomic assignment approach using Kraken 2 for the patient's samples.**

| | True positive | True negative | False positive | False negative |
|---|---|---|---|---|
| Sample 2 – *Campylobacter* spp. | 2 | 8 | 2 | 0 |
| Sample 4 – *Clostridioides difficile* | 1 | 7 | 4 | 0 |
| Sample 5 – *Campylobacter* spp. | 2 | 7 | 3 | 0 |
| Sample 6 – *Campylobacter* spp. | 2 | 6 | 4 | 0 |
| Sample 11 – *Escherichia coli* (Shiga toxin-producing) | 1 | 7 | 4 | 0 |
| Sample 12 – *Salmonella* spp. | 1 | 6 | 5 | 0 |
| Sample 13 – *Escherichia coli* (Shiga toxin-producing) | 1 | 6 | 5 | 0 |
| Sample 15 – *Salmonella* spp. | 1 | 6 | 5 | 0 |
| Sample 10 – Giardia intestinalis | 1 | 7 | 4 | 0 |
| Sample 14 – *Cryptosporidium* spp. | 1 | 6 | 5 | 0 |
| Sample 3 – Human mastadenovirus F | 1 | 5 | 6 | 0 |
| Sample 7 – Human mastadenovirus F | 1 | 7 | 4 | 0 |
| **Total** | **15** | **78** | **51** | **0** |

transposons associated to various pathogens (S3 Table). For *G. intestinalis*, PCR results were obtained for all samples in the 10⁻¹ dilution, while negative PCR results were obtained for the 10⁻⁵ dilution. In the 10⁻³ dilution, PCR results were obtained for all replicates of BP1, and for one replicate of BP2, while the other BP samples were negative. *G. intestinalis* was identified with on average 8 reads in the triplicates of only one of the samples (BP1), and was not detected in any other samples, nor in the negative controls.

**Table 3. Average number of sequencing reads versus PCR Cq values for dilutions of the spiked faecal BP\* samples, BP1, BP2, BP4 and BP5, and the corresponding non-spiked BP_Neg samples.**

| Agent | Dilution | $10^{-1}$ | $10^{-3}$ | $10^{-5}$ | Non-spiked sample | $10^{-1}$ | $10^{-3}$ | $10^{-5}$ | Non-spiked sample |
|---|---|---|---|---|---|---|---|---|---|
| | Sample No. | Average No. of Reads (%) | | | No. of Reads** (%) | Average Cq values | | | |
| *Campylobacter jejuni* | BP1 | 20,604.3 (0.14) | 11,085.0 (0.062) | 9,811.7 (0.058) | 8,599.0 (0.064) | 24.8 | 31.4 | 39.0 | Neg |
| | BP2 | 11,150.7 (0.08) | 2,086.3 (0.018) | 1,818.3 (0.017) | 2,066.0 (0.017) | 24.7 | 31.5 | 37.6 | Neg |
| | BP4 | 22,062.0 (0.11) | 4,053.3 (0.019) | 3,587.0 (0.023) | 4,860.0 (0.028) | 24.0 | 30.8 | 37.7 | Neg |
| | BP5 | 8,775.7 (0.08) | 141.0 (0.001) | 70.0 (0.0004) | 338.0 (0.0019) | 25.5 | 31.9 | 38.1 | Neg |
| | **Mean±SD** | **15,648.2 ±6,829.4** | **4,341.4 ±4,329.3** | **3,821.8 ±3,869.5** | **3,965.8 ±3,607.2** | **24.8 ±0.6** | **31.4 ±0.5** | **38.1 ±0.6** | **NA\*\*\*** |
| *Giardia intestinalis* | BP1 | 8.0 (0.000055) | 0.0 | 0.0 | 0.0 | 29.1 | 36.0 | Neg | Neg |
| | BP2 | 0.0 | 0.0 | 0.0 | 0.0 | 29.6 | 37.8**** | Neg | Neg |
| | BP4 | 0.0 | 0.0 | 0.0 | 0.0 | 34.4 | 0 | Neg | Neg |
| | BP5 | 0.0 | 0.0 | 0.0 | 0.0 | 33.7 | 0 | Neg | Neg |
| | **Mean±SD** | **2 ±4.7** | **0.0** | **0.0** | **0.0** | **31.7 ±2.7** | **18.5 ±21.3** | **NA** | **NA** |
| Human Mastadenovirus F | BP1 | 39,825.3 (0.28) | 363.0 (0.0020) | 0.0 | 0 | 19.5 | 25.5 | 32.9 | 35.8 |
| | BP2 | 34,040.0 (0.26) | 345.0 (0.0029) | 0.0 | 0 | 18.1 | 25.3 | 32.9 | 33.8 |
| | BP4 | 31,688.3 (0.16) | 296.3 (0.0014) | 0.0 | 0 | 17.8 | 25.5 | 31.1 | 31.6 |
| | BP5 | 17,673.0 (0.16) | 100.6 (0.0008) | 0.0 | 280.0 (0.0016) | 19.0 | 26.5 | 32.3 | 33.4 |
| | **Mean±SD** | **30,806.7 ±12,587.9** | **276.3 ±180.5** | **0.0** | **70.0 ±140.0** | **18.6 ±0.8** | **25.7 ±0.5** | **32.3 ±0.9** | **33.7 ±1.7** |

\*BP: Faecal samples from four healthy donors, denoted BP1, BP2, BP4 and BP5

\*\*Results of BP-Neg based on one technical replicate

\*\*\*NA Not applicable

\*\*\*\*Cq value based on results from one replicate

## Detection of target pathogens by taxonomic assignment of Metagenome-assembled genomes (MAGs)

**Clinical samples.** All twelve clinical samples were individually assembled and binned, producing a total of 298 MAGs, with median 25 MAGs per sample. The bacterial samples were taxonomically assigned using the Genome Taxonomy database Toolkit (GTDb-Tk), and bacterial pathogens were found in four samples (Table 4). A *C. coli* MAG of 189 contigs, with 58.9% completeness and 853,379 basepairs, and no contamination was found in Sample 6. In Sample 11, an *E. coli* MAG (closest placement was uropathogenic *E. coli* strain UMN026) with 461 contigs, 56.9% completeness, 2,399,059 bp and no contamination was detected. In Sample 12, *S. enterica* was detected having 504 contigs, 51.7% completeness, 4,315,987 bp and 1.72% contamination, while in Sample 13 *E. coli* (closest placement was *Shigella flexnerii*) with 119 contigs, 98.97% completeness, 4,649,863 bp and contamination of 0.28% was found. Additional taxonomic classification of the bins by Kraken 2, reported *C. coli* for Sample 6 and *Salmonella enterica* for Sample 12. For both Samples 11 and 13, the bins were classified as *E. coli*. For samples 2, 4, 5 and 15, the target bacterial pathogen was not detected. The viral samples were taxonomically assigned using geNomad, identifying the target viral pathogen adenovirus in clinical samples 3 and 7. Results from checkV however suggested misassemblies by Megahit, as indicated by larger than expected genome sizes and high numbers of host genes in adenovirus contigs. Viral samples were therefore additionally assembled using Haploflow, which resulted in higher quality viral assemblies. For sample 3, this resulted in a MAG consisting of 12 contigs with completeness of 36.8% and size of 12,965 bp (Table 4). For sample 7, analyses recovered 3 contigs with 100% completeness, a total of 35,362 bp and no contamination. Both these samples had closest taxonomic assignment to Human adenovirus 41. For clinical samples infected with parasites, there were not enough species-specific reads to generate MAGs.

**Table 4. Classification and characteristics of metagenome-assembled genomes from clinical and spiked faecal samples.**

| Sample No. | MAG No. | Taxonomy classification | Genome accession | Average nucleotide[1]/amino acid[2] identity (%) | Alignment fraction[3] | Genome size (bp) | GC content (%) | Completeness (%) | Contamination (%) | No. of contigs |
|---|---|---|---|---|---|---|---|---|---|---|
| **Clinical samples** | | | | | | | | | | |
| **6** | 16 | *Campylobacter coli* | GCF_000254135.1 | 98.48 | 0.97 | 853,379 | 32.8 | 58.91 | 0.00 | 189 |
| **11** | 39 | *Escherichia coli* | GCF_000026325.1 | 97.64 | 0.90 | 2,399,059 | 50.58 | 56.90 | 0.00 | 461 |
| **12** | 7 | *Salmonella enterica* | GCF_000006945.2 | 98.54 | 0.89 | 4,315,987 | 52.7 | 51.72 | 1.72 | 504 |
| **13** | 19 | *Escherichia coli* | GCF_002950215.1 | 97.88 | 0.85 | 4,649,863 | 50.77 | 98.97 | 0.28 | 119 |
| **3** | 85 | Human adenovirus 41 | ON442328.1 | 99.1 | 74.07 | 12,965 | 52.19 | 36.8 | 0 | 12 |
| **7** | 10 | Human adenovirus 41 | OP174917.1 | 99.9 | 95.5 | 35,362 | 51.22 | 100 | 0 | 3 |
| **Spiked samples** | | | | | | | | | | |
| **Co-assembly of BPs 10⁻¹** | Bin_175 | *Campylobacter jejuni* | GCF_001457695.1 | 97.5 | 0.92 | 1,766,442 | 31.0 | 75.02 | 0.23 | 165 |
| **BP1–10⁻¹a** | 50 | Human mastadenovirus F | ON815883.1 | 99.92 | 98.2 | 33,879 | 51.05 | 96.4 | 0 | 4 |
| **BP1–10⁻¹b** | 35 | Human mastadenovirus F | ON815883.1 | 99.92 | 98.78 | 33,835 | 51.04 | 96.2 | 0 | 4 |
| **BP1–10⁻¹c** | 25 | Human mastadenovirus F | ON815883.1 | 99.9 | 98.7 | 33,916 | 51.06 | 96.5 | 0 | 4 |
| **BP2–10⁻¹a** | 13 | Human mastadenovirus F | ON815883.1 | 99.92 | 98.07 | 33,715 | 51.01 | 95.9 | 0 | 5 |
| **BP2–10⁻¹b** | 6 | Human mastadenovirus F | ON815883.1 | 99.92 | 95.28 | 33,889 | 51.06 | 96.4 | 0 | 4 |
| **BP2–10⁻¹c** | 48 | Human mastadenovirus F | ON815883.1 | 99.92 | 97.96 | 33,655 | 50.97 | 95.8 | 0 | 6 |
| **BP4–10⁻¹a** | 362 | Human adenovirus 41 | OP174917.1 | 99.9 | 99.24 | 33,988 | 51.05 | 96.7 | 0 | 3 |
| **BP4–10⁻¹b** | 3114 | Human adenovirus 41 | OP174917.1 | 99.9 | 100 | 33,970 | 51.04 | 96.6 | 0 | 2 |
| **BP4–10⁻¹c** | 2218 | Human adenovirus 41 | OP174917.1 | 99.73 | 92.37 | 33,993 | 51.04 | 96.7 | 0 | 2 |
| **BP5–10⁻¹a** | 58 | Human mastadenovirus F | ON815883.1 | 99.91 | 84.13 | 33,529 | 50.99 | 95.3 | 0 | 9 |
| **BP5–10⁻¹b** | 422 | Human mastadenovirus F | ON815883.1 | 99.77 | 80.78 | 23,769 | 49.52 | 69.4 | 0 | 13 |

*(Continued)*

**Table 4.** (Continued)

| Sample No. | MAG No. | Taxonomy classification | Genome accession | Average nucleotide[1]/amino acid[2] identity (%) | Alignment fraction[3] | Genome size (bp) | GC content (%) | Completeness (%) | Contamination (%) | No. of contigs |
|---|---|---|---|---|---|---|---|---|---|---|
| **BP5–10⁻¹c** | 319 | Human mastadenovirus F | MK962808.1 | 99.1 | 74.07 | 14,547 | 47.34 | 42.6 | 0 | 9 |

[1]Average nucleotide identity is given for bacterial samples

[2]Average amino acid identity is given for viral samples

[3]Alignment fraction indicates the alignment fraction between the query and reference genome

**Spiked samples.** Coassembly and binning of the $10^{-1}$ dilution replicates from BP1, BP2, BP4 and BP5 resulted in a total of 186 MAGs. This included a MAG (bin_175), taxonomically assigned as *C. jejuni* with 165 contigs, 75% completeness, 1,766,442 bp and 0.23% contamination (Table 4). In this dilution the coverage was on average 1.70 ± 0.45 (S4 Table). At the $10^{-3}$ dilution the average coverage of the MAG dropped to 0.13 ± 0.16 and was similar to the $10^{-5}$ dilutions (0.13 ± 0.24). The average coverage in BP_neg (non-spiked) samples was 0.007 ± 0.004.

Adenovirus MAGs were identified in all $10^{-1}$ dilution replicates from BP1, BP2, BP4 and BP5 separately as well as in the $10^{-1}$ coassembly. As for the clinical samples, results from checkV suggested misassemblies by Megahit, and all $10^{-1}$ spiked samples were thus additionally assembled using Haploflow. The taxonomically assigned adenovirus MAGs, which were recovered from all $10^{-1}$ spiked samples, consisted of 2–13 contigs, with median genome size of 33,857 bp and 42.6–96.7% completeness (Table 4). Here the coverage was on average 950.81 ± 304.23 (S5 Table). For both *C. jejuni* and HAdVF the classification results were confirmed by taxonomic classification of the bins by Kraken 2.

Due to the low number of total reads assigned to *G. intestinalis* in the spiked samples, no *Giardia* MAGs were generated.

**Species-specific detection of *Giardia* spp.** From clinical sample 10, only 410 reads mapped to the *Giardia lamblia (intestinalis)* reference genome GCA_000182665.1 and 367 reads to *G. intestinalis* GCA_000002435.2 (S6 Table). The spiked samples had <70 reads mapping to either *Giardia* spp. genome with most reads mapping to *G. intestinalis* (GCA_000002435.2). Because of low number of reads for *Giardia*, it was not possible to perform assembly.

**Detection of virulence-associated genes in bacterial clinical samples.** Virulence-associated genes were predicted only for clinical faecal samples diagnosed with bacterial pathogens, based on either metagenomic assemblies and/or MAGs (Table 5). In Sample 6, diagnosed with *Campylobacter* spp. by PCR, the adherence genes *cadF* and *pebA,* as well as the motility gene *flg* were found in the same contig in the metagenome assembly. These genes were also found in the *C. coli* MAG. For Sample 4 (*C. difficile*), several virulence genes were detected, however none specifically related to *C. difficile* virulence. In Sample 11 (STEC) the intimin gene, *eae*, and other genes related to the Locus of enterocyte effacement (LEE) as well as genes associated to Type III secretion system (TTSS) were detected. Also the enterohemolysin gene, *hlyA*, was detected. Sample 13 (STEC) contained Shiga toxin 1 genes (*stx1*A and *stx*1B), genes associated to TTSS and hemolysin genes (*hly*A to *hly*D genes). For both samples 11 and 13 we also identified virulence genes commonly found in extraintestinal pathogenic *E. coli* in several contigs in the assemblies. In the MAGs, only two genes (*nleD, nleH*) in TTSS were identified in Sample 11, while for Sample 13, the *E. coli* MAG included the genes associated to Shiga toxin 1. Sample 12 (*Salmonella* spp.) contained various virulence genes associated to *Salmonella* spp., including TTSS genes, salmonella plasmid virulence locus genes (*spv*) and invasion genes (*invA-invH*) associated to Salmonella pathogenicity island 1, *sopB* associated to Salmonella pathogenicity island 5 and adherence genes (*fimA–fimC, fimE, fimI*). In addition, *Salmonella* pathogenicity island genes *pefA&B* were identified. In the *S. enterica* MAG of Sample 12, the *spv* genes were not identified, but the *inv* and other genes associated to TTSS as well as *fim, lpf* and *sopB* were identified.

**Table 5. Virulence genes associated to detected GI pathogens in assemblies and in MAGs.**

| Sample No | Pathogen detected | In assemblies | In MAGs |
|---|---|---|---|
| 2 | *Campylobacter* spp. | ND[1] | ND |
| 5 | *Campylobacter* spp. | ND | ND |
| 6 | *Campylobacter* spp. | *cadF, pebA, flg, chu, motA, fli, cheY, katA,* | *cadF, pebA, flg, motA, fli* |
| 4 | *Clostridioides difficile* | ND | ND |
| 11 | *Escherichia coli* (STEC) | *eae,* genes in LEE[2] and TTSS[3], *hlyA* | *nleD, nleH* (both in TTSS) |
| 13 | *Escherichia coli* (STEC) | *stx1A, stx1B,* genes in TTSS, *hlyA-hlyD* | *stx1A, stx1,* genes in TTSS |
| 12 | *Salmonella* spp. | Genes in TTSS (e.g., *spv, inv*), *sopB, fim, lpf, iroDEN, sodCI, csg* | Genes in TTSS (*inv*) *sopB, fim, lpf, csg* |
| 15 | *Salmonella* spp. | ND | ND |

[1]Not detected

[2]LEE: Locus of enterocyte effacement

[3]TTSS: Type three-secretion system

## Discussion

In this study we evaluated metagenomic shotgun sequencing in comparison to PCR as a diagnostic tool for detection of gastrointestinal pathogens in a clinical hospital laboratory. In total 55 samples were sequenced, including clinical faecal samples (n = 12), spiked faecal samples (n = 36) and control samples (n = 7), and different bioinformatic approaches were tested for detection of pathogens and virulence genes.

Faeces is a complex sample material, thus requiring a relatively high sequencing depth to detect and characterise pathogens as well as potential virulence and resistance genes present in the sample material [43,44]. In this study Non-pareil was used to assess the sequencing coverage of samples, which indicated sufficient sequencing quality to cover the diversity of the datasets.

One of the main draws of clinical metagenomics is the possibility of detecting different types of microbial pathogens in a single test. However, it is well known that nucleic acid extraction is a major challenge and introduces bias, as different microorganisms require vastly different conditions for efficient extraction. In this study we used a DNA extraction protocol specifically developed for faecal samples [13], and treated and extracted all samples uniformly. From our results, this protocol appeared to be efficient for extraction of both the included Gram-negative and Gram-positive bacterial pathogens, as well as for adenovirus. Only a few reads were however assigned to parasites, both in the clinical and spiked samples, indicating low efficacy and/or low quality of DNA extraction. Thus, it might be beneficial to implement nucleic acid extraction protocols more specifically targeted at parasites to increase sensitivity of shotgun sequencing of such samples [45–47]. Detection of parasites however appeared to be highly specific, as no other samples contained any reads associated to parasites. Initially, after arrival to the lab, the faecal samples in this study were stored at 4°C while routine diagnostics were performed, before storage at −20°C. Although short-term storage at 4°C is not supposed to alter the microbiota composition to a great extent [48], we did not control for overgrowth or loss of certain microbial species in our material during the storage period. In routine diagnostics, it has been common to enrich the sample for certain pathogens like *Salmonella* spp., however this was not performed in this study. *Salmonella* spp. were detected by Kraken 2 in both PCR-positive clinical samples. However, due to a small sample size in this study, it is difficult to estimate whether lack of enrichment might have affected the results or not.

One of the major challenges of clinical metagenomics, given the unbiased sequencing of any nucleic acid present in the sample, is the potential detection of contaminants stemming from either sample collection, nucleic acid extraction, library preparation or sequencing [49–51]. Only in recent years has it become standard practice to include negative controls to account for this problem [52]. In this study, three negative controls were included for DNA extraction and sequencing. Although the number of reads detected in the low complexity negative controls cannot be directly compared

to those of high complexity faecal samples, the controls can be used to indicate potential sources of contamination. Reads assigned to *E. coli* were for instance detected in the negative controls with Lysis buffer and Phosphate buffered saline. As no such reads were observed in the Molecular grade water, most probably these reads originated from the reagents used to make the Lysis buffer and PBS. However, contamination from the DNA extraction kit or from laboratory procedures cannot be excluded [53–55].

**Detection of bacterial pathogens**

Kraken 2 is a commonly used tool for taxonomic assignment in metagenomics studies [21,56–58], which in general detected all pathogenic species included in this study. However, other potential gastrointestinal pathogens were also detected in most of the samples, often in high abundances. Colonization with *C. difficile* in healthy humans is reported to range between 0–17% [59–61], while non-pathogenic *E. coli* are part of the gastrointestinal microbiome in approximately 90% of human individuals [62]. Here, a closer inspection of the *C. difficile* reads observed in the BP_Neg samples revealed misclassification by Kraken 2 and Kraken Uniq, most probably caused by presence of mobile genetic elements (S3 Table), suggesting that more stringent mapping and better curated databases are required to conclusively identify *C. difficile*-specific reads in metagenomes when using Kraken 2 and KrakenUniq. In the spiked samples, *C. jejuni* was the only pathogen that was detected in all dilutions by taxonomic assignment of reads. Furthermore, while the non-spiked control samples (BP_Neg) were negative for *C. jejuni* by PCR, *C. jejuni* reads were detected in these samples. Again, further analysis suggested misclassified reads due to non-specific sequences of *Campylobacter* spp. or related species present in the Kraken database. Thus, although not tested in this study, implementing more relevant and updated databases/tools for detection of targets that are pathogen-specific, will enhance specificity as previously reported [57]. Additionally, while not part of the current study, long-read sequencing – which is well-known for enhancing the assembly of mobile genetic elements and repetitive regions – could have facilitated better categorization of misclassified reads [63]. Only four (50%) of the clinical pathogenic species were detected in MAGs (Table 4), in which also pathogen-specific virulence genes were detected. These observations were most probably due to higher numbers of reads present from these specific pathogens in the samples. For the remaining samples, too low read numbers to generate good assemblies, and incomplete binning of contigs might explain why taxonomic assignment of MAGs, as well as detection of virulence genes were not possible [43,44]. Implementing additional assembly and binning tools here could potentially enhance the results [64,65]. The *E. coli* bins in Sample 11 and 13, which showed closest taxonomic assignment to uropathogenic *E. coli* and *Shigella flexnerii*, respectively, were most likely reflecting incomplete or incorrect binning of contigs due to the heterogenous nature of *E. coli* and/or the presence of more *E. coli* genomes in the samples. We furthermore observed a discrepancy in the presence of some virulence genes in the metagenomic contigs as compared to MAGs (Table 5). These results might suggest that the virulence genes could be present on contigs and/or mobile genetic elements that are not correctly binned into MAGs [66]. While metagenomics sequencing has the potential to detect genes from all organisms in a sample, it might be difficult to evaluate which pathogens are relevant for the observed symptoms by a solely taxonomic approach, because detection of certain pathogens requires identification of specific virulence genes [57,67,68]. For example, while generic *E. coli* was detected by Kraken 2, it is necessary to use additional tools and databases such as the Virulence Factor Database for specific detection of the virulence genes *eae* and/or *stx* genes essential for identification of STEC. Also for detection of toxigenic *C. difficile*, detected by the presence of the toxin genes *tcdA* and/or *tcdB*, analysis with Virulence factor database was nessecary [67,69]. In this study, *stx1* genes were not detected in Sample 11, neither in assemblies nor MAGs, most probably due to the low amount of Shiga toxin 1 gene in the sample, as indicated by a high Cq value by PCR.

For the spiked samples, a Campylobacter MAG was generated using co-assembled reads from the $10^{-1}$ dilution samples, whereas no MAGs were generated with further dilutions. Low coverage and a low number of reads mapping to the Campylobacter MAG however suggested that the concentration of the pathogen was too low for assembly and binning in the $10^{-3}$ and $10^{-5}$ dilutions.

## Detection of HAdV-F

While HAdV-F was identified with a high number of reads in the two PCR-positive clinical samples, neither HAdV-F nor any other adenoviruses were detected in any of the remaining clinical samples, indicating a specific detection of the virus using the Kraken approach. The PCR assay used to detect HAdV-F in the spiked samples in this study, is designed to detect human mastadenoviruses within the serotypes A-G. Our results showed that all spiked samples, as well as the BP_Neg samples that were not spiked, were PCR-positive whereas the non-template control was negative (Table 3). These results are most probably reflecting that DNA from adenovirus serotypes other than HAdV-F were also present in the faecal samples. The Cq values observed in the $10^{-5}$ dilution were only slightly lower as compared to the BP_Neg samples, indicating only a small amount of HAdV-F DNA present in the $10^{-5}$ dilution. Corresponding results were however not observed by sequencing, with no assigned reads to any adenoviruses in three out of four non-spiked BP_Neg samples, as well as in the $10^{-5}$ dilutions (Table 3). Thus, the Kraken approach appears to have lower sensitivity than PCR, although higher specificity. In the non-spiked BP5_Neg sample, we observed more reads for HAdV-F than in the $10^{-3}$ diluted sample BP5-3c (S2 Table). These two samples were positioned adjacent to each other during library preparation and we cannot exclude sample contamination between plate wells or that the samples might have been accidentally switched.

The large number of MAGs generated in the clinical HAdV-F samples and the $10^{-1}$ dilution of the spiked samples by *de novo* assembly using Megahit resulted in a large number of short contigs and very large genomes. A closer inspection revealed misassembled hybrid bacterial and viral contigs produced by the assembler (Megahit) as the most plausible reason. Contrasting our initial difficulties, the second approach of assembling using Haploflow did however produce more complete adenovirus MAGs from these samples (Table 4). Thus, our results highlight potential pitfalls and indicate that different bioinformatics strategies are needed for de-novo assembly of viral metagenomic reads as compared to bacterial reads.

## Detection of *Giardia intestinalis*

In contrast to the other spiked pathogens, *C. jejuni* and HAdV-F, only a few reads were taxonomically assigned to *G. intestinalis* for both the clinical and the spiked samples by Kraken (Table 1b, Table 3). We therefore investigated a species-specific mapping approach to detect the parasite, which showed that few reads mapped to the *Giardia* spp. reference genomes (S6 Table). Furthermore, results from PCR analyses showed high or negative Cq values for *G. intestinalis* already in the $10^{-3}$ dilution, and negative results for the $10^{-5}$ dilution. Low amount of parasites and/or inefficient extraction of parasite DNA as previously discussed might be the cause of both the PCR and sequencing results, although the sensitivity of the PCR approach is clearly higher than sequencing for this organism.

Due to financial constraints, we were only able to include 12 clinical samples in this study. Consequently, each pathogen was represented only once or twice in the sample material. Furthermore, the study should ideally have included both RNA and DNA viruses. However, only HAdV, a DNA virus, was represented. Despite these limitations, the included clinical and spiked samples represent significant gastrointestinal pathogens of bacterial, viral, and parasitic origin. Our results thus highlight various factors influencing metagenomics analysis of these diverse pathogen groups.

Although metagenomics sequencing of faecal samples has a definite potential for pathogen detection in certain situations compared to more limited PCR panels, the method still has some limitations affecting implementation into routine diagnostics. The data presented in this study indicate that different bioinformatics tools and strategies are required for detection of the broad range of different pathogens that might be present in faecal samples. Furthermore, the turn-around time and costs are still high compared to PCR, although pricing of metagenomics sequencing is slowly decreasing [57]. In recent years, more diagnostic metagenomics studies of faecal samples have been published [3,4,7,57]. There is however a need for standardisation and validation studies to ensure that methods, results and interpretations are comparable and reproducible.

## Conclusion

In summary, metagenomics sequencing of the faecal samples in this study show that all included gastrointestinal pathogens were detected using various bioinformatics approaches, yet with lower sensitivity as compared to PCR. Background microbiome and introduced kitome remains challenges, some of which could potentially be alleviated by strict use of quality controls and curated databases. Despite hurdles like lower sensitivity, as well as higher cost and labour compared to PCR, metagenomics analysis has the potential to detect novel or unexpected pathogens and add comprehensive information about the pathogens. Thus, clinical metagenomic sequencing of faecal samples is a promising diagnostic tool.

## Supporting information

**S1 Fig. Read statistics.** Read statistics are shown after quality control filtering for clinical samples, spiked samples, and negative controls.
(TIF)

**S1 Table. Nonpareil diversity parameters of clinical and spiked samples.** For spiked samples, average values for each sample and dilution were calculated.
(XLSX)

**S2 Table. Taxonomic assignement of reads based on results from Kraken 2 and Bracken.**
(XLSX)v

**S3 Table. BLAST results of mobile genetic elements.**
(XLSX)

**S4 Table. Anvio-profile statistics for coverage, total mapped reads and total detection of the *C. jejuni* MAG for dilutions of the spiked faecal samples, and the corresponding non-spiked BP_Neg samples.**
(XLSX)

**S5 Table. Anvio-profile statistics for coverage, total mapped reads and total detection of the HAdV-F MAG for dilutions of the spiked faecal samples, and the corresponding non-spiked BP_Neg samples.**
(XLSX)

**S6 Table. Average number of reads from clinical and spiked faecal samples mapping to *Giardia* spp. reference genomes.**
(DOCX)

## Acknowledgments

The authors acknowledge the Genomics Core Facility (GCF), Norwegian University of Science and Technology (NTNU) Whole genome sequencing in the study.

## Author contributions

**Conceptualization:** Kjersti Haugum, Anuradha Ravi, Jan Egil Afset, Christina Gabrielsen Ås.

**Data curation:** Kjersti Haugum, Anuradha Ravi, Christina Gabrielsen Ås.

**Formal analysis:** Kjersti Haugum, Anuradha Ravi, Christina Gabrielsen Ås.

**Funding acquisition:** Jan Egil Afset.

**Investigation:** Jan Egil Afset.

**Methodology:** Kjersti Haugum, Anuradha Ravi, Jan Egil Afset, Christina Gabrielsen Ås.

**Project administration:** Kjersti Haugum.

**Software:** Kjersti Haugum, Anuradha Ravi, Christina Gabrielsen Ås.

**Supervision:** Christina Gabrielsen Ås.

**Validation:** Jan Egil Afset, Christina Gabrielsen Ås.

**Visualization:** Anuradha Ravi.

**Writing – original draft:** Kjersti Haugum, Jan Egil Afset, Christina Gabrielsen Ås.

**Writing – review & editing:** Kjersti Haugum, Anuradha Ravi, Jan Egil Afset, Christina Gabrielsen Ås.

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
