## [Decision Letter · Decision Letter 0]

14 Jun 2025

PONE-D-25-08885Evaluation of shotgun metagenomics as a diagnostic tool for infectious gastroenteritisPLOS ONE

Dear Dr. Haugum,

Thank you for submitting your manuscript to PLOS ONE. After careful consideration, we feel that it has merit but does not fully meet PLOS ONE’s publication criteria as it currently stands. Therefore, we invite you to submit a revised version of the manuscript that addresses the points raised during the review process.

We look forward to receiving your revised manuscript.

Kind regards,

Adriana Calderaro

Academic Editor

PLOS ONE

Journal Requirements:

Reviewers' comments:

Reviewer's Responses to Questions

**Comments to the Author**

1. Is the manuscript technically sound, and do the data support the conclusions?

Reviewer #1: Yes

2. Has the statistical analysis been performed appropriately and rigorously? 

Reviewer #1: N/A

3. Have the authors made all data underlying the findings in their manuscript fully available?

Reviewer #1: Yes

4. Is the manuscript presented in an intelligible fashion and written in standard English?

Reviewer #1: Yes

5. Review Comments to the Author

Reviewer #1: In this manuscript, the author compared the applicability of shotgun metagenomic and PCR for the diagnosis of infectious gastroenteritis. This is very relevant and will help improve diagnostic protocols and routines. The experimental setup appears reasonable, and the authors present their findings effectively. There are some points that I would like to discuss before making any decision.

• In this work, misclassification is closely linked to mobile genetic elements, which may be one reason. However, it is not the only one. Another possible reason may be the microbial abundance in metagenomics. Regardless of the reason, there is insufficient evidence in this manuscript to support mobile genetic elements as a cause of misclassification, and I believe this assertion is too strong to be made.

• The authors emphasize that bioinformatics analyses could impact the metagenomic results. This is true. However, I believe the manuscript does not provide sufficient evidence to support this claim. For instance, both the assembler and the binner could have a significant impact. In this manuscript, the authors chose to test only Megahit and Metabat2. It has been shown that both assembler and binner behaviour differ when they are used for low and high coverage data. So, preferring one over the other is very challenging in metagenomics. Indeed, many publications have suggested a multi-tool approach. Additionally, there are many different taxonomy tools; however, only Kraken2 and its associated tools were tested. Thus, although the statement is correct, the evidence is insufficient.

• I believe the manuscript will benefit from constructing a confusion matrix that includes both the false positive and negative rates for sequencing as a potential diagnostic method compared to the traditional routine method.

• What is missing from the discussion is the applicability of long-read sequencing to overcome the limitations in the field of metagenomic assembly, especially when it comes to mobile genetic element and their repetitive regions.

Please provide a percentage of unclassified reads for Kraken.

Please provide the percentage of unique mapped reads, depth of coverage, and breadth of coverage after mapping the reads back to assemblies (some of these values are provided but not in percentages).

The authors declared that they mapped the sequencing reads against the host genome where any actual maps were removed. This sounds like a correct approach; however, the authors did not specify the percentage of the host genome contamination.

L 415 / L 421: final spiked concentration is contrasting; please double-check.

L 440: One of the discussed points in this manuscript was the DNA extraction technique and its effectiveness. While I agree with the authors, unfortunately, the current work only presents DNA concentration and A260/A280 ratios. It would be nice to show the A260/A230 as well.

L 493 and L502: any reason to only use Megahit and Metabat2? It is hard to prefer one over the other, specifically for assemblers. Some assemblers performed better for low coverage data.

L 495: Why are authors considered co-assembly?

L 503: At this stage, it is uncertain whether the authors considered all the MAGs for downstream analyses or only selected medium and high-quality MAGs. Please also provide statistics on the quality of the MAGs.

L 508: Why did the authors suddenly decide to use GTDb-Tk for the taxonomy classification of MAGs?

L 526: What was the justification to shift to SPAdes for assembly?

L 527: How is percentage coverage calculated? The method is not consistent across the literature, so please provide more details.

6. PLOS authors have the option to publish the peer review history of their article (what does this mean? ). If published, this will include your full peer review and any attached files.

**Do you want your identity to be public for this peer review?** For information about this choice, including consent withdrawal, please see our Privacy Policy .

Reviewer #1: No

---

## [Author Response · Author response to Decision Letter 1]

27 Jun 2025

Haugum et al. - Response to Reviewers

Manuscript number: PONE-D-25-08885

Title: Evaluation of shotgun metagenomics as a diagnostic tool for infectious gastroenteritis

We thank Reviewer #1 for reviewing and commenting on our manuscript “Evaluation of shotgun metagenomics as a diagnostic tool for infectious gastroenteritis”. We have now worked through the manuscript and have, to the best of our knowledge, taken the comments from the Reviewer into consideration. Please see below for details.

Comment to Journal Requirements:

Response to comment 5: We recognize PLOS ONE’s policy regarding the use of the phrase “data not shown”. In this instance, the phrase 'data not shown' was somewhat redundant. It was included to indicate that referencing these specific results was unnecessary, as the outcomes from the second assembly approach using Haploflow are presented in Table 3. As such, in our opinion, the phrase can be removed while the rest of the text remain unchanged. The phrase “data not shown” is removed from line 531.

Review Comments to the Author

Reviewer #1:

• In this work, misclassification is closely linked to mobile genetic elements, which may be one reason. However, it is not the only one. Another possible reason may be the microbial abundance in metagenomics. Regardless of the reason, there is insufficient evidence in this manuscript to support mobile genetic elements as a cause of misclassification, and I believe this assertion is too strong to be made.

Response to reviewer: We thank the reviewer for this comment, and recognize that that other reasons for misclassification exist.

We have now included S3 Table in the manuscript. In this table, we show results from BLAST searches of the ten largest contigs assembled from C. jejuni and C. difficile reads from each of the four non-spiked BP negative samples (i.e. donor faeces without pathogens added). The results showed that the majority of the BLAST hits are related to mobile genetic elements and we therefore suggest there is evidence that mobile genetic elements may contribute to misclassification. Furthermore, we have rephrased the text to clarify, please see lines 333-335, and lines 478-481.

• The authors emphasize that bioinformatics analyses could impact the metagenomic results. This is true. However, I believe the manuscript does not provide sufficient evidence to support this claim. For instance, both the assembler and the binner could have a significant impact. In this manuscript, the authors chose to test only Megahit and Metabat2. It has been shown that both assembler and binner behaviour differ when they are used for low and high coverage data. So, preferring one over the other is very challenging in metagenomics. Indeed, many publications have suggested a multi-tool approach. Additionally, there are many different taxonomy tools; however, only Kraken2 and its associated tools were tested. Thus, although the statement is correct, the evidence is insufficient.

Response to reviewer: We thank reviewer for the comment, and do agree on noted points. We did not aim to extensively test many bioinformatics tools in this study. The tools were chosen for their suitability to our data, as demonstrated by their performance in peer-reviewed literature. We have now added more references to reflect this (line 189 and 199). We have in addition changed the text to apply to comment regarding use of Kraken, please see line 479.

• I believe the manuscript will benefit from constructing a confusion matrix that includes both the false positive and negative rates for sequencing as a potential diagnostic method compared to the traditional routine method.

Response to reviewer: We thank the reviewer for the comment and have now included results for the suggested confusion matrix as Table 2. Please also see lines 294-296 for further information.

• What is missing from the discussion is the applicability of long-read sequencing to overcome the limitations in the field of metagenomic assembly, especially when it comes to mobile genetic element and their repetitive regions.

Response to reviewer: We appreciate the reviewer's comment and agree with the noted point. We have implemented a sentence reflecting this; please see lines 481-485.

• Please provide a percentage of unclassified reads for Kraken.

Response to reviewer: Information regarding unclassified reads is now included in the results, lines 267-269, and in the S2 Table.

• Please provide the percentage of unique mapped reads, depth of coverage, and breadth of coverage after mapping the reads back to assemblies (some of these values are provided but not in percentages).

Response to reviewer: We have now added percentage of unique mapped reads to the S4 Table. In the same table, depth of coverage and breath of coverage are in addition included.

• The authors declared that they mapped the sequencing reads against the host genome where any actual maps were removed. This sounds like a correct approach; however, the authors did not specify the percentage of the host genome contamination.

Response to reviewer: Information regarding human contamination is now included in the results, lines 267-269.

• L 415 / L 421: final spiked concentration is contrasting; please double-check.

Response to reviewer: We recognize that the text in these two sentences can be confusing. In line 415 (now line 112) we refer to the starting concentration of 2.0 x 108 CFU/mL. This is diluted in 1 ml with 100 µL, giving the final concentration of 2.0 x 107 CFU/mL (line 421, now line 117). We have rewritten the text to clarify this, please see line 117-118.

• L 440: One of the discussed points in this manuscript was the DNA extraction technique and its effectiveness. While I agree with the authors, unfortunately, the current work only presents DNA concentration and A260/A280 ratios. It would be nice to show the A260/A230 as well.

Response to reviewer: Thank you for commenting on this. A260/A230 was also measured, and revised text is now included in line 137.

• L 493 and L502 (L 189 and 199): any reason to only use Megahit and Metabat2? It is hard to prefer one over the other, specifically for assemblers. Some assemblers performed better for low coverage data.

Response to reviewer: In this study, we selected the assembler and binner that were best suited for our data, based on their performance as reported in peer-reviewed literature as mentioned above. We agree with the reviewer that different assemblers and binning tools perform differently. However, as the aim was not to compare or benchmark many bioinformatics tools, more tools were not used. While using additional or different assemblers/binners might have provided more comprehensive data, we have addressed this in lines 189 and 199, with adding more references for the tools. Please also see line 491 in the manuscript.

• L 495 (L 191): Why are authors considered co-assembly?

Response to reviewer: We thank reviewer for this comment. In this study, co-assembly was performed to improve genome assembly, by increasing read depth and thus aiming at improving the recovery of low-abundance species, as well as to generate genomes as finished/complete as possible. Thereafter, reads from each sample were individually mapped to quantify completeness and coverage of these reads. To avoid confusion, we have added a sentence regarding this in the manuscript; please see line 191-192.

• L 503 (L 200): At this stage, it is uncertain whether the authors considered all the MAGs for downstream analyses or only selected medium and high-quality MAGs. Please also provide statistics on the quality of the MAGs.

Response to reviewer: Thank you for the comment. Here all bacterial MAGs (line 200) were imported into anvi´o, and result with statistics are provided in Table 4. For virus MAGs, no further work was done. If any quality statistics are missing, we would be happy to provide this information upon request.

• L 508 (L 204): Why did the authors suddenly decide to use GTDb-Tk for the taxonomy classification of MAGs?

Response to reviewer: We thank the reviewer for the comment. Our initial strategy to use Kraken for taxonomix assignment was based on the use of reads to classify pathogens (and other microbes). However, Genome Taxonomy database Toolkit (GTDb-Tk) is a commonly used tool for taxonomic assignment after assembly and binning. Additionally, we used Kraken for taxonomic assignment on contigs.

• L 526 (L222): What was the justification to shift to SPAdes for assembly?

Response to reviewer: All reads were first assembled with Metabat. Due to very few reads for Giardia spp. using this approach, we changed to the species-specific approach for Giardia detection, where reads that mapped to Giardia were extracted and assembled with SPAdes. The reason for using SPADes is that it is widely recognized in the literature as one of the best assemblers for reads derived at the species level.

• L 527 (L 223): How is percentage coverage calculated? The method is not consistent across the literature, so please provide more details.

Response to reviewer: We thank the reviewer for making us aware of this. “Percent coverage” is changed to “alignment length” based on BLAST search in the text, please see line 223.

---

## [Decision Letter · Decision Letter 1]

14 Aug 2025

Evaluation of shotgun metagenomics as a diagnostic tool for infectious gastroenteritis

PONE-D-25-08885R1

Dear Dr. Haugum,

We’re pleased to inform you that your manuscript has been judged scientifically suitable for publication and will be formally accepted for publication once it meets all outstanding technical requirements.

Kind regards,

Adriana Calderaro

Academic Editor

PLOS ONE

Additional Editor Comments (optional):

Reviewers' comments:

Reviewer's Responses to Questions

**Comments to the Author**

1. If the authors have adequately addressed your comments raised in a previous round of review and you feel that this manuscript is now acceptable for publication, you may indicate that here to bypass the “Comments to the Author” section, enter your conflict of interest statement in the “Confidential to Editor” section, and submit your "Accept" recommendation.

Reviewer #2: All comments have been addressed

2. Is the manuscript technically sound, and do the data support the conclusions?

Reviewer #2: Yes

3. Has the statistical analysis been performed appropriately and rigorously? 

Reviewer #2: Yes

4. Have the authors made all data underlying the findings in their manuscript fully available?

Reviewer #2: Yes

5. Is the manuscript presented in an intelligible fashion and written in standard English?

Reviewer #2: Yes

6. Review Comments to the Author

Reviewer #2: The authors have satisfactorily addressed all reviewer comments in the revised manuscript. I therefore recommend the manuscript for acceptance.

7. PLOS authors have the option to publish the peer review history of their article (what does this mean? ). If published, this will include your full peer review and any attached files.

**Do you want your identity to be public for this peer review?** For information about this choice, including consent withdrawal, please see our Privacy Policy .

Reviewer #2: No

---

## [Editor Report · Acceptance letter]

PONE-D-25-08885R1

PLOS ONE

Dear Dr. Haugum,

I'm pleased to inform you that your manuscript has been deemed suitable for publication in PLOS ONE. Congratulations! Your manuscript is now being handed over to our production team.

Kind regards,

on behalf of

MD, PhD, Full Professor Adriana Calderaro

Academic Editor

PLOS ONE